# Adolescent Aggressive Riding Behavior: An Application of the Theory of Planned Behavior and the Prototype Willingness Model

**DOI:** 10.3390/bs13040309

**Published:** 2023-04-04

**Authors:** Sheng Zhao, Xinyu Chen, Jianrong Liu, Weiming Liu

**Affiliations:** School of Civil Engineering and Transportation, South China University of Technology, Guangzhou 510641, China

**Keywords:** psychological factors, integrated model, riding, aggressive behaviors

## Abstract

Cycling has always been popular in China, especially during the years when the government encouraged green travel. Many people participate in rides to ease traffic congestion and increase transfer convenience. Due to the disorganized and tidal nature of cycling, cyclists create many conflicts with other groups. Adolescents are vulnerable road users with a strong curiosity and risk-taking mindset. Identifying the factors influencing adolescents’ aggressive riding behavior can assist in developing strategies to prevent this behavior. An online questionnaire was used to collect data on bicycling among students in a middle school in Guangzhou, China. The theory of planned behavior (TPB) and prototype willingness model (PWM) have been applied to study travel behavior and adolescent risk behavior. To investigate the impact of psychological variables on adolescent aggressive behavior, we used TPB, PWM, TPB + PWM, and an integrated model. Behavioral intentions are greatly influenced by attitudes, subjective norms, and perceived behavioral control. Both descriptive and moral norms played a role in behavioral willingness. The integrated model explained 18.3% more behavioral variance than the TPB model. The social reactive pathway explained more variance in behavior than the rational path.

## 1. Introduction

Bike riding is a convenient and economical option for many commuters, and it is also a good choice for short trips. The lack of physical shell protection, however, exposes bicyclists to greater risks and environmental pollution than drivers [1]. The majority of bicycle-related crashes result from road facilities (e.g., cycle paths and intersection types), other road users’ faults, and the violation by the rider themselves [2,3,4,5,6]. Although the first two types of factors are uncontrollable, psychological characteristics may influence cyclists’ violations. Studies have examined the influence of psychological factors on pedestrian violations and driving violations, but few studies have focused on risky riding [7,8,9,10]. Adolescents are psychologically immature and risk-taking, easily influenced by peers, and often involved in risky behaviors [11]. In 2019, riders under the age of 18 accounted for 8.4% of bike-sharing trips in China, indicating that teens like to ride [12]. This study aims to analyze the psychological factors contributing to teenagers’ risky riding behavior and proposes effective measures to improve teenagers’ safety on the road.

### 1.1. Background

In the past five years, bicycle sales have increased substantially in China, and some citizens have changed their commuting habits. Several cities have set up bike lanes, green lanes, and special signals to encourage residents to travel by bicycle [13]. Guangzhou promoted the use of unlocked shared bicycles in September 2016, and due to inadequate bicycle infrastructure, the city has added bike lanes to improve traffic [14]. Initially, urban roads were designed for vehicular traffic, so bicycles and motor vehicles shared the same road. The city has added bike lanes and bike signals to the roadway infrastructure to improve safety for cyclists. Greenways in Guangzhou cover a distance of 3500 km, improving the convenience and satisfaction of its residents. To attract more people to use public transportation, we need to push for the promotion of cycling and make it the “last mile” connection [15]. The younger generation is the driving force behind bicycle riding, and cultivating their travel habits is beneficial to the sustainable development of transportation.

It is important to regulate the behavior of cyclists as they are not as strictly supervised as drivers, causing many conflicts on the road. Thousands of Chinese cyclists lose their lives in traffic accidents every year, and the cumulative number of deaths exceeded 20,000 as of 2017 [16]. A total of 93% of road fatalities, as reported by the World Health Organization, take place in low- and middle-income nations. Young people are especially vulnerable to traffic collisions, with about 330 young people dying in traffic collisions each day [17]. From 2010 to 2013, there was a decrease of 4% in fatalities due to prevention activities.

Adolescents are adventurous and do not take safety seriously, resulting in more traffic injuries [18,19]. Lack of parental monitoring makes teens more likely to engage in risky behavior [20]. Several studies have also linked adolescent risk behaviors to independence or peer influence [21]. Students’ risky travel behaviors pose a threat to their safety, including jostling on the streets, using cell phones, and riding bikes without helmets [22,23,24]. Instead of improving transportation facilities, it would be more beneficial to help young people regulate their travel behavior. Scholars found psychological factors associated with pedestrian violations, including perceived behavioral control, prototype perception, and behavioral willingness [25]. The level of risk perception also affects the driver’s behavior in taking preventive measures [26]. Psychological factors contribute significantly to risk-taking behaviors. It is plausible to believe that psychological factors may have a role in risky riding behavior based on the findings of previous studies.

### 1.2. Factors Influencing Risky Riding Behaviors

Many studies have been conducted on risky riding behaviors, including analysis of individual characteristics, questionnaires, and psychological factors. The influence of individual characteristics on behavior is mainly reflected in age and gender. Younger and older cyclists are at higher risk of traffic conflicts, while other age groups will adopt more protective behaviors and perceive risks more keenly [27,28]. Men report more anger and are more likely to be involved in a collision on a ride than women [29,30].

Researchers have used questionnaires to analyze the risky riding behavior of various groups, including the cyclist behavior scales and the psychological factors scales. Scholars examined risky riding behavior among youth, aged 14–25, using the Cyclist Behavior Questionnaire (CBQ) [31]. Study [32] established the cyclist anger scale (CAS), which measures cyclists’ risky interactions with other road users. The Cycling Anger Expression Inventory (CAX) quantifies cyclists’ aggressive behavior, and people with higher levels of anger show more aggressive behavior while riding [33]. The most common aggressive behaviors among drivers and cyclists are verbal expressions and physical actions, such as verbal abuse and hitting others [34].

Risky behaviors are associated with social learning theory and psychological factors. An individual’s interaction with the social environment can influence the decision to engage in risky behavior. For example, riders are prone to aggressive behavior when they lack subjective risk, feel stressed, or have been drinking [35,36]. In social learning theory, aggressive behavior is a result of the norms and patterns to which individuals are exposed. People are prone to aggressive behaviors under the influence of family members, the culture of the family, and social media [37]. This phenomenon can be explained by prototype preferences, where the higher an individual’s approval of another person’s behavior, the more they likely they are to engage in such behaviors.

Some researchers have demonstrated that psychological factors have an impact on risky riding behavior. Psychological factors associated with risk behavior include attitudes, individual norms, risk perception, and prototype preferences [38,39,40]. High-minded riders exhibit less aggressive behavior [41]. Researchers found that risk comparisons, attitudes toward alcohol in traffic, and normative factors accounted for 23% of risky riding intentions [18]. The combination of the cyclist anger scale and the psychological factors scale was effective in explaining the risk-taking behavior of cyclists. In the current academic literature, aggressive riding behavior in adolescents has not been analyzed by exploring the influence of psychological factors. The study examined factors that could contribute to adolescents’ aggressive riding behavior by combining these two scales. To examine the influence route of variables, we employ both the theory of planned behavior (TPB) and the prototype willingness model (PWM).

### 1.3. Model Framework

#### 1.3.1. Theory of Planned Behavior

The TPB model investigates the role of intentions in behavior by examining the impact of attitude, subjective norms, and perceived behavioral control on behavioral intentions [42]. Attitude (ATT) refers to an individual’s perception of a behavior, and is determined by their salient beliefs [43]. Subjective norms (SN) involve social pressure on behavior and indicate whether individuals should act [44]. Perceived behavioral control (PBC) consists of perceived control, perceived confidence, and perceived difficulty. Perceived difficulty alone is not sufficient to measure PBC [45]. Furthermore, it has been found that perceived difficulty predicts intentions and behaviors better than perceived control [46].

Scholars found that image, attitudes, and normativity had significant effects on the behavior of female adolescents riding to school using the TPB model [47]. The TPB also explained the intentions of 47% of cyclists to run red lights and 65% of cyclists to turn left at intersections [40]. Scholars have found that positive attitudes are associated with cyclists’ intentions to commute by bike [48]. To better explain the behavior, researchers added other psychological variables, such as descriptive norms and moral norms, to the TPB model, which is called the extended TPB model [49,50]. Descriptive norms (DN) refer to individuals’ perceptions of others’ behavior [49]. When most people engage in a behavior, individuals may also have the intention to engage in such behavior. Moral norms (MN) are considered to be an individual’s view of the rightness or wrongness of engaging in a certain behavior [42]. This factor has been used to analyze the effects of risky behaviors, such as riding while using a cell phone and drunk-driving [50,51].

#### 1.3.2. Prototype Willingness Model

PWM is based on TPB and adds prototype perception (prototype favorability (PF) and prototype similarity (PS)) and behavioral willingness (BW), but does not include perceived behavioral control. PWM has two influence paths, including the rational path and the social reactive pathway. The rational path is similar to the TPB model, attitude and subjective norms affect intention and then affect behavior. The social reactive pathway means that attitude, subjective norms, and prototype perceptions affect willingness, and then affect intention and behavior. Prototype perceptions indicate whether an individual perceives a prototype behavior as positive or negative. An individual is more likely to adopt a behavior if they perceive the prototype as favorable and similar to their own [52]. The concept of behavioral willingness differs from the concept of behavioral intention. In contrast to an intention, which is a person’s future planned conduct, a behavioral willingness is a person’s willingness to execute certain behaviors in a particular event or environment [53]. The PWM identifies two types of influence paths: factors that affect intentions, which affect behaviors, called the rational path, and factors that affect willingness, which affects intentions and behaviors, called the social reactive pathway [54]. PWM is predominantly applicable to studies of adolescent violations, such as smoking behavior, alcohol abuse, and violations during the COVID-19 lockdown [55,56,57]. This study will validate the advantages of PWM in explaining aggressive riding behavior in adolescents.

#### 1.3.3. Integrated Model

Some scholars combine the factors of TPB and PWM to analyze behavior, called the integrated model [58,59]. The influence path of the integrated model is to add perceived behavioral control, descriptive norms, and moral norms to PWM, and additionally, analyze the influence of these variables on intention and willingness. Several studies have validated the validity of this model, which has demonstrated a significant improvement in the explained variance of behavior. However, a study on pedestrian behavior found that the PWM model had better predictive power than the integrated model [25]. The purpose of this study is to analyze the differences between TPB, PWM, TPB + PWM, and the integrated model in explaining adolescents’ aggressive riding behavior.

### 1.4. The Current Study

Due to the wide applicability and explanatory power of the TPB and PWM models in predicting behavior, the structure of these two models was considered in this study. Studies have utilized these two models to analyze the use of cell phones while cycling, running a red light, speeding behavior, etc., [9,60,61]. PWM and integration models (including the factors of TPB and PWM) explain the behavior better than TPB [59]. Currently, fewer studies have investigated aggressive riding behavior among youths. Identifying the factors that influence youth road safety is critical to improving their safety on the road. The following hypotheses were put forward in this study:Attitudes, subjective norms, and perceived behavioral control are significantly associated with behavioral intentions;Prototype perceptions (prototype favorability and prototype similarity) are related to behavioral willingness;Behavioral intention and willingness are associated with aggressive riding behaviors.

## 2. Methodology

### 2.1. Study Area and Participant

The survey was conducted at Peiying Middle School in Xiguan, Liwan District, Guangzhou. According to a previous study of youth travel behavior, more students at this school commute by bicycle. The roads near the Xiguan Peiying Middle School are flat and suitable for teenagers to ride. A total of 87% of high school students travel less than 4 km from home to school, a distance most parents support their children to ride [62]. Cycling lanes and bike signals are available in the area, and the riding environment is conducive to youth cycling.

The questionnaire was approved by teachers and students who were informed that it was intended only for academic research purposes. We sent a link to parents through the Questionnaire Star website, and the students completed the questionnaire with the assistance of their parents. There were 510 questionnaires returned between April and May, 2022, of which 463 were valid. According to the survey, the age range of the participants was 12–20 years old, with 68.3% being 15–19 years old and 83.5% being in high school. The ratio of males to females was more balanced, with 237 female students (58.1%). The student’s families have a good standard of living, with 94.2% of them having an income above average. A total of 83.6% of students rode more than three times a week, and 294 rode more than five times. It is feasible to analyze aggressive riding behavior at this school since many students ride bicycles every week.

### 2.2. Questionnaire Design

We refer to study [30] for risky riding behavior, which is a questionnaire in which cyclists self-report eight common behaviors categorized as risky riding and aggressive riding. The following six behaviors were selected as aggressive behaviors: weaving in and out of traffic, speeding, running red lights, going against the flow of traffic, making rude gestures to others or verbally abusing them, and honking loudly to move others out of the way. Participants were asked to report how often they had recently engaged in these behaviors on a five-point Likert scale (1 for “never” and 5 for “always”).

#### 2.2.1. The Factors of TPB

The scale of potential variables in the TPB model includes six kinds of aggressive riding behaviors for teenagers. On a five-point Likert scale, participants were asked to rate their agreement with the items (1 for “strongly agree” and 5 for “strongly disagree”). The attitude refers to [25], describing participants’ perceptions of aggressive behavior, such as “I believe it is immoral to ride a bicycle while still running a red light”. The subjective norms refer to [63], describing how supportive people are of the participant’s behavior, such as “My friends and family do not want me to ring the bell loudly when cycling”. Perceived behavioral control reflects students’ ability to control their behavior in various situations, for example, “I can control myself when riding a bike without rampaging”. Behavioral intentions indicate the cyclist’s intention for six behavior patterns, such as “When riding a bicycle in the future, I will certainly not shuttle back and forth to overtake”.

#### 2.2.2. The Factors of PWM

The items of attitude, subjective norm, and behavioral intention in the PWM model are the same as those in the TPB model. Other latent variables also include six aggressive riding behaviors, and participants need to answer the degree of identity on the five-point Likert scale (1 for “strongly agree” and 5 for “strongly disagree”). Prototype favorability refers to [64], describing participants’ cognition of others’ behaviors, such as “I think it is reckless to hinder others from overtaking when riding a bike”. Prototype similarity indicates how similar the student’s behavior is to other riders, for example, “How similar is it to a person who accelerates into an intersection when the signal changes from yellow to red?”. Prototype perception is a reactive decision-making process that affects behavior through willingness [65]. Behavioral willingness indicates the rider’s willingness to perform a certain behavior, referring to [54], for example, “How willing are you to ring the bell loudly when pedestrians slowly walk in front of you?”.

#### 2.2.3. Additional Factors

Descriptive norms and moral norms are added to the integrated model, and the design of problems includes the above six aggressive riding behaviors. Descriptive norms refer to [49], and one of the items is “ The frequency of loud ringing of bells when people ride bicycles ”. Participants answered on a five-point Likert scale (1 for “never” and 5 for “always”). Moral norms refer to [50], and one of the items is “ If I run a red light while riding a bicycle, I feel very annoyed afterwards”. Participants responded on a five-point Likert scale (1 for “strongly agree” and 5 for “strongly disagree”).

## 3. Results

### 3.1. Measurement Model

The confirmatory factor analysis (CFA) measures the link between latent variables and their corresponding items, as well as the validity of the questionnaire. Table 1 shows the convergent validity of each latent variable. All factor loadings (0.609 to 0.879) greater than 0.6 indicate a reasonable structure [66]. The composite reliability (CR) and Cronbach’s alpha are greater than 0.7, indicating a high reliability of the construct and a reasonable scale design [67]. The average variance extracted (AVE) value should be more than 0.5 and range from 0.527 to 0.664 for all variables [68].

Discriminant validity is the relationship between comparing AVE values and structural covariance. The lower triangle reflects the correlation between the latent variables, while the value on the diagonal is the square root of AVE [68]. The diagonal values in Table 2 are higher than the other values in the same row and column, supporting the discriminant analysis’s validity.

### 3.2. Structural Equation Modeling

This study examines four models, including the basic TPB and PWM models, the TPB+PWM models, and the integrated model that includes descriptive norms and moral norms. Table 3 displays the index criteria of the structural equation model. The chi-square freedom ratio ($\chi^{2}$/DF) should be less than three, and the CFI and TLI should be larger than 0.9 [69,70]. The model is suitable since both the standardized root mean square residual (SRMR) and root mean square error of approximation (RMSEA) values are less than 0.08 [69]. The fitting indices of the four models all reached the standards, which indicates that these models were valid.

### 3.3. Comparison of Models

In contrast to the other two models, TPB + PWM and the integrated model both increased the explained variance of the intention, as seen in Table 4. The integrated model significantly improves the explained variance of behavioral willingness by 12.3% higher PWM and 8.9% higher TPB + PWM, respectively. In all three of these models, the explained variance of behavior is increased by approximately 18% over TPB. The factors in TPB only affect behavior through the rational path, whereas the social reactive pathway is also included in the last three models. The results indicate that the social reflective pathway explained 18% of the behavior, suggesting that the aggressive riding behavior of adolescents is primarily a result of the stress response. In a survey of teenagers’ future non-smoking behavior, the social reactive pathway also explains 16% of the behavioral variance [55]. Overall, the results show that the integrated model provides the best explanation of the behavior, followed by the PWM and TPE + PWM. The pathways of action of the factors in the integrated model will be analyzed in detail.

### 3.4. The Pathway Analysis of the Integrated Model

The integrated model explained 43.2% of behavioral intentions, 37.1% of behavioral willingness, and 31.4% of aggressive behaviors. The integrated model includes the structure of TPB and PWM, as well as the effect of descriptive and moral norms on actions. The specific path analysis is illustrated in Figure 1. Dashed lines indicate that the parameter is nonsignificant at the 95% confidence level, whereas solid lines indicate that the parameter is significant at the 95% confidence level. Attitudes (β = 0.338, p<0.001), subjective norms (β = 0.177, p<0.001), perceived behavioral control (β = 0.365, p<0.001), and behavioral willingness (β = −0.084, p<0.05) had significant effects on behavioral intentions in this model. Additionally, the more likely participants were to show a willingness to engage in aggressive behaviors, the more they would inhibit plans to not engage in such behaviors. Perceived behavioral control (β = −0.247, p<0.01), perceived similarity (β = 0.309, p<0.001), descriptive norms (β = 0.420, p<0.001), and moral norms (β = −0.173, p<0.05) play an important role in behavioral willingness. A higher sense of moral responsibility and control over aggressive behavior will result in a lower willingness to engage in aggressive behavior. Intention (β = −0.194, p<0.001) and willingness (β = 0.324, p<0.001) had a significant impact on aggressive behaviors, while perceived behavioral control had no direct effect.

The analysis of the direct and indirect effects on the integrated model is shown in Table 5. There was a significant direct effect of attitudes, subjective norms, and perceived behavioral control on intentions. Prototype similarity and behavioral willingness had large indirect impacts on intentions. On actual behavior, there were substantial indirect impacts of descriptive norms, perceived behavioral control, moral norms, and prototype similarity.

## 4. Discussion

Cycling is a convenient way to travel short distances. In China, the government has encouraged the development of the bike-sharing market to urge citizens to travel more sustainably. The World Health Organization recommends that secondary school students exercise for one hour daily. Bicycle commuting can contribute to students’ exercise and health. This study analyzes the influence of psychological factors on adolescents’ aggressive cycling behavior and suggests strategies for improving safety.

As in previous studies [71,72], attitudes, subjective norms, and perceived behavioral control significantly affected behavioral intentions. Positive attitude disincentives aggressive behavior and future policies are more likely to influence the behavior of riders who have a negative attitude. Schools can, for example, conduct civilized activities, standardize students’ daily communication, and prevent verbal attacks.

Several studies have examined the role of subjective norms. The subjective normative role of friends was most significant in adolescents’ cycling behavior [73], while the effect on cell phone use while cycling was weak [61]. The effect of subjective norms on aggressive riding behavior was slightly weaker but still significant. Increasing public awareness of risky behavior may reduce aggressive behavior in students. Unlike other aggressive behaviors [74], perceived behavioral control significantly affects adolescent riding. Adolescents who have more control over aggressive behavior are less likely to engage in such behavior and maintain a safe traffic order on their rides. Educational intervention is an effective measure to prevent aggressive behavior [75].

The descriptive norms in this study refer to the frequency of behaviors such as yelling, speeding, and running red lights while riding a bicycle by family members and friends. Descriptive norms have an important influence on behavioral intentions. A similar effect of this factor was found in the intention to travel by bicycle and public transport among Swedish residents [76]. Regulating parental riding behavior may have an impact on student behavior. The behavior of parents and peers is widely believed to have a direct or indirect influence on adolescents’ decisions [77]. The same is true for prototype similarity, where students that are more supportive of aggressive behaviors are more likely to behave similarly themselves.

Morality tends to discourage people from taking risks and it is an important part of Chinese culture. The moral norms of college students influence their environmental behavior [78], as well as their habit of texting while driving [79]. In addition to subject education, schools should place more emphasis on developing students’ sense of responsibility. Governments may also promote ethical concepts through the display of billboards in public places such as subway stations, bus stops, and communities.

The social response path explained behavioral variance more than the rational path in this study. Two structures are involved in the social response pathway: perception of the risk prototype and openness to taking risks [54]. Furthermore, it can be argued that adolescents engage in risky behavior based on their willingness to act rather than a deliberate decision. The result is similar to that of smoking [55], alcoholism [56], and cyberbullying [80] in adolescents. Therefore, aggressive riding behavior among adolescents may be considered voluntary. Changing traffic conditions does little to improve students’ behaviors; making them aware of the dangers is a better way to change their behavior.

## 5. Conclusions and Limitations

In this study, the fitting results of the four models were compared, and the model incorporating the PWM framework performed better than TPB in prediction. Path analysis was conducted using the integrated model with the highest explained variance. Attitudes, subjective norms, and perceived behavioral control all have a major impact on intentions. Perceived behavioral control, prototypical similarity, descriptive norms, and moral norms are important predictors of behavioral willingness. Descriptive norms and ethical norms differ from previous findings [50,51] in that they affect willingness rather than intention in this study. It can be determined that willingness is a more important factor than intention for aggressive cycling behaviors. The integrated model explained 18.3% more of the variance than TPB, indicating that the social reactive pathway explains behavior more. Analyzing aggressive riding behaviors in adolescents through the integrated model was demonstrated to be feasible.

There are also some limitations to this study. High school students constituted the majority of the survey population, but student behavior varies with age. The behavior of younger students may be more reckless [81,82], which will be investigated. Due to fear of parental or teacher concerns, participants may have underreported their behavior, which can lead to a bias in the results. A one-week collection of videos of actual behaviors can be conducted with a small number of volunteers.

## Figures and Tables

**Figure 1 behavsci-13-00309-f001:**
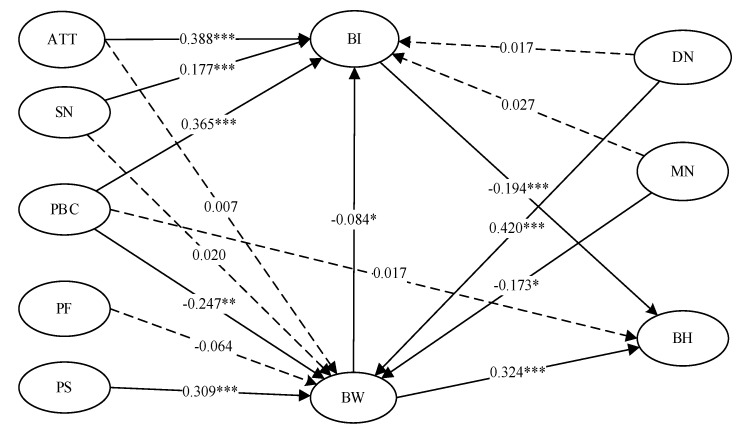
The pathway of the integrated model, * *p* < 0.05, ** *p* < 0.01, *** *p* < 0.001.

**Table 1 behavsci-13-00309-t001:** The reliability and convergence validity of the latent variables.

Constructs	Standard Loading	Cronbach’s α	CR	AVE
Attitudes	0.662–0.788	0.868	0.870	0.527
Subjective norms	0.770–0.857	0.922	0.922	0.664
Descriptive norms	0.639–0.833	0.900	0.900	0.602
Perceived behavioral control	0.724–0.778	0.888	0.889	0.572
Prototype favorability	0.625–0.827	0.891	0.895	0.589
Prototype similarity	0.711–0.879	0.919	0.921	0.660
Behavioral intentions	0.733–0.827	0.902	0.903	0.607
Behavioral willingness	0.609–0.820	0.864	0.869	0.527
Moral norms	0.748–0.795	0.897	0.897	0.593
Behaviors	0.649–0.798	0.880	0.881	0.554

**Table 2 behavsci-13-00309-t002:** The construct of discriminant validity.

	1	2	3	4	5	6	7	8	9	10
1	0.726									
2	0.547	0.815								
3	−0.014	−0.007	0.776							
4	0.479	0.317	−0.133	0.756						
5	0.361	0.283	−0.066	0.337	0.767					
6	−0.136	−0.095	0.294	−0.211	−0.159	0.812				
7	0.545	0.440	−0.101	0.541	0.330	−0.211	0.779			
8	−0.216	−0.142	0.405	−0.299	−0.212	0.454	−0.362	0.726		
9	0.539	0.371	−0.119	0.430	0.282	−0.181	0.400	−0.312	0.770	
10	−0.198	−0.111	0.279	−0.268	−0.268	0.354	−0.353	0.510	−0.205	0.744

Note: 1 = attitudes, 2 = subjective norms, 3 = descriptive norms, 4 = perceived behavioral control, 5 = prototype favorability, 6 = prototype similarity, 7 = behavioral intentions, 8 = behavioral willingness, 9 = moral norms, 10 = behaviors.

**Table 3 behavsci-13-00309-t003:** The goodness-of-fit indices.

Fit Index	Criterion	TPB	PWM	TPB + PWM	The Integrated Model
$\chi^{2}$		588.138	1373.637	1661.178	2462.961
DF		397	804	1058	1673
$\chi^{2}$/DF	1 < $\chi^{2}$/DF < 3	1.481	1.709	1.570	1.472
CFI	>0.9	0.975	0.950	0.953	0.952
TLI	>0.9	0.973	0.947	0.950	0.949
RMSEA	<0.08	0.032	0.039	0.035	0.032
SRMR	<0.08	0.040	0.049	0.046	0.045

**Table 4 behavsci-13-00309-t004:** Explanatory power for each model (%).

Models	Behavioral Intentions	Behavioral Willingness	Behaviors
TPB	41.9		13.1
PWM	36.3	24.8	31.1
TPB + PWM	43.1	29.4	31.1
Integrated model	43.2	37.1	31.4

**Table 5 behavsci-13-00309-t005:** Direct and indirect effects of the integrated model.

	Behavioral Intention			Behavior		
	Direct	Indirect	Total	Direct	Indirect	Total
Attitudes	0.388 *** (0.000)	−0.001 (0.957)	0.386 *** (0.000)	-	−0.073 (0.135)	−0.073 (0.135)
Subjective norms	0.177 ** (0.001)	−0.002 (0.782)	0.175 ** (0.001)	-	−0.028 (0.310)	−0.028 (0.310)
Descriptive norms	0.017 (0.776)	−0.035 (0.054)	−0.018 (0.741)	-	0.140 *** (0.000)	0.140 *** (0.000)
Perceived behavioral control	0.365 *** (0.000)	0.021 (0.082)	0.386 *** (0.000)	0.017 (0.778)	−0.155 *** (0.000)	−0.138 *** (0.015)
Moral norms	0.027	0.014	0.041	-	−0.064 *	−0.064 *
	(0.617)	(0.116)	(0.435)		(0.016)	(0.016)
Prototype favoraiblity	-	0.005 (0.285)	0.005 (0.285)	-	−0.022 (0.210)	−0.022 (0.210)
Prototype similarity	-	−0.026 * (0.048)	−0.026 * (0.048)	-	0.105 *** (0.000)	0.105 *** (0.000)

Notes: *** *p* < 0.001, ** *p* < 0.01, * *p* < 0.05. The *p* value is indicated in parentheses.

## Data Availability

The dataset used during the current study is available from the corresponding author upon reasonable request.

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
