# Peer review of "Adolescent Aggressive Riding Behavior: An Application of the Theory of Planned Behavior and the Prototype Willingness Model"

_behavsci, 2023, doi:10.3390/bs13040309_

Round 1
Reviewer 1 Report
In this study different theoretical models are tested to explain students’ riding-related behaviors in China. The first part of the manuscript is well written and a rationale for the study is built by presenting data on riding associated risks in China. Nevertheless, the theoretical models that are tested with respect to this behavior are not well presented. This part should be improved. Also the measures section needs some major changes. Please, consider the following points.
p.3, when introduced, the TPB and PWM models should be described (they are described later in the text)
p.3. rather than describing how the paper is articulated and organized, I suggest presenting some hypotheses based on the models that are going to be tested.
When the TPB is described, a more detailed definition of subjective norms should be provided
The authors state that “The TPB has been developed to include several psychological variables, such as descriptive norms and 156 moral norms”; nevertheless, moral norms were not present in Ajzen’s proposal (1991); their additional role has been tested through subsequent studies. It would be useful to provide a definition for the two kinds of norms.
The Prototype willingness model is not well described; all its components and the link between them should be better identified and defined. The same can be said for the Integrated model; how does it integrate the two previous model?
The measures section should be improved. It is not clear how each factor has been assessed (how many items? Is the scale reliable?). The authors seem to report another definition of each construct, rather than explain how it was measured. The difference between descriptive, subjective and moral norms should not be reported here, but previously.
Results. Given that some models are compared, also the Akaike information criterion (AIC) should be reported.
It is said that “The pathway also explains 16% of the variance in 261 future non-smoking behavior of adolescents”. Do the authors refer to another study? This should be made clearer.
Reviewer 2 Report
I miss psychological social learning theories to explain aggressive and risky driving. Dangerous driving behaviour may be influenced by a number of factors: 'the young male syndrome' and observational learning may contribute to the choice to drive or respond aggressively on the roads. it could promote a more precise modelling of dangerous driving behaviours. Though evolutionary psychological theory can make an important contribution to the understanding of risky driving behaviour, societal influences, such as the presence of others and individual difference variables, interact with competitiveness and can influence the decision to engage in risk-taking behaviours. According to social interaction theory, aggressive behaviour is social influence behaviour, whereby coercive acts are used to obtain things of value, bring about retributive justice, or promote social and self-identities.Further suggested articles to include into the article, mostly in theoretical part:
DRIVER AGGRESSION: THE ROLE OF PERSONALITY, SOCIAL CHARACTERISTICS, RISK AND MOTIVATION E.M. Grey, T.J. Triggs and N.L. Haworth Monash University Accident Research Centre, 1989
Singhal, D., & Wiesenthal, D. (2021). Evolutionary psychology and dangerous driving behaviour. In T. K. Shackelford (Ed.), The SAGE handbook of evolutionary psychology: Applications of evolutionary psychology (pp. 374–397). Sage
Round 2
Reviewer 1 Report
The authors have answered to the points I raised in my previous review. I have some other suggestions to improve this revised version.
-I suggest not starting sentences with a citation in brackets, as it not facilitates the reading
-Please, replace “rivers” with “riders”
-the sentence “Descriptive norms (DN) express an individual's belief in the behavior of others” should be revised
-The concept of “moral norms” should be better described as well
-Please replace “behavior intention” with “behavioral intention”
-Please avoid causal terms (e.g., in the hypotheses, as the methodology adopted in this study does not allow to infer causal links between variables
-English should be carefully revised
